# WD Repeat and HMG Box DNA Binding Protein 1: An Oncoprotein at the Hub of Tumorigenesis and a Novel Therapeutic Target

**DOI:** 10.3390/ijms241512494

**Published:** 2023-08-06

**Authors:** Zhiwei Zhang, Qing Zhu

**Affiliations:** Division of Abdominal Tumor Multimodality Treatment, Cancer Center, West China Hospital, Sichuan University, No. 37 Guoxue Alley, Chengdu 610041, China; 2021224020151@stu.scu.edu.cn

**Keywords:** WD repeat and HMG-box DNA binding protein 1, proliferation, DNA damage, tumorigenesis

## Abstract

WD repeat and HMG-box DNA binding protein 1 (WDHD1) is a highly conserved gene from yeast to humans. It actively participates in DNA replication, playing a crucial role in DNA damage repair and the cell cycle, contributing to centromere formation and sister chromosome segregation. Notably, several studies have implicated WDHD1 in the development and progression of diverse tumor types, including esophageal carcinoma, pulmonary carcinoma, and breast carcinoma. Additionally, the inhibitor of WDHD1 has been found to enhance radiation sensitivity, improve drug resistance, and significantly decrease tumor cell proliferation. This comprehensive review aims to provide an overview of the molecular structure, biological functions, and regulatory mechanisms of WDHD1 in tumors, thereby establishing a foundation for future investigations and potential clinical applications of WDHD1.

## 1. Introduction

The WD repeat and HMG-box DNA binding protein 1 (WDHD1), also known as acidic nucleoplasmic DNA-binding protein 1 (AND-1), was initially discovered through genetic screens aimed at identifying genes involved in chromosomes transmission during cell division [1,2]. In the model organism *Saccharomyces cerevisiae* (*budding yeast*), the human counterpart of WDHD1 is referred to as chromosome transmission fidelity factor 4 (CTF4) [3].

WDHD1 is a large protein comprising 1129 amino acids and has a molecular weight of 126 kilodaltons (kDa). It contains several distinct domains that contribute to its biological function, including a WD40 repeat domain at the N terminus, a Sep B domain, and an HMG domain at the C terminus [4]. Proteins containing the WD40 repeat domain have been found to facilitate protein–protein interactions and can function as connectors or regulators in various cellular processes [5]. Additionally, WDHD1 contains the highly conserved SepB domain, which plays a crucial role in DNA replication processes. The HMG domains are DNA-binding motifs found in the HMGB protein group [6].

Recent research has provided evidence supporting the involvement of WDHD1 in regulating various pathophysiological processes. CTF4 has been recognized as a key player in DNA replication, DNA damage repair, sister chromatin cohesion, and ensuring accurate chromosome transmission [7,8,9]. Similarly, WDHD1 plays crucial roles in DNA replication [10,11,12], cell cycle progression [13,14], sister chromatid cohesion [15,16], DNA damage [17], and embryonic development [18]. Moreover, WDHD1 has demonstrated its importance in various types of cancers, including lung cancer [19], cholangiocarcinoma [20], cervical cancer [21], and breast cancer [22]. Collectively, WDHD1 contributes to cancer progression and metastasis through its impact on biological functions and downstream molecules. As a novel therapeutic target of cancer, WDHD1 knockdown was also associated with cisplatin resistance [23,24] and radiosensitivity [25].

However, there is still a need for a comprehensive investigation into the underlying molecular mechanisms linking WDHD1 to diverse tumors. Therefore, this review aims to provide an overview of current research on the function and molecular mechanisms of WDHD1 over the past decades. Furthermore, this paper presents possible approaches for understanding the governing mechanisms and exploring the therapeutic implications of WDHD1 in associated cancers, thus setting the stage for future investigations.

## 2. Evolutionary Conservation of WDHD1

WDHD1 is a replisome component protein that is widely conserved from yeast to humans [26]. The N terminus WD40 repeat domain and the conserved SepB region of WDHD1 have been identified in orthologous sequences across a comprehensive set of 101 species (TreeFam: http://www.treefam.org/, accessed on 6 July 2023). To analyze the evolutionary conservation of WDHD1, a phylogenetic tree analysis was conducted using sequence data obtained from the National Center for Biotechnology Information (NCBI) database (https://www.ncbi.nlm.nih.gov/, accessed on 22 June 2023) by Molecular Evolutionary Genetics Analysis (MEGA) 11 [27]. The results of the phylogenetic tree analysis demonstrate the evolutionary conservation of WDHD1 across various species, including Mus musculus, Rattus norvegicus, Xenopus laevis, Danio rerio, and Homo sapiens (Figure 1A). Additionally, a query of the human genome carried out via the GEO database (https://www.ncbi.nlm.nih.gov/geo/, accessed on 22 June 2023) revealed that WDHD1 is located on the opposite strand of chromosome 14, specifically at 55,088,816–54,877,237 (Figure 1B). Furthermore, it was observed that WDHD1 is expressed in a wide range of human tissues and organs.

## 3. The Molecular Structure of WDHD1

The human WDHD1 is considered the homologous gene of the budding yeast CTF4. In terms of primary structure, the budding yeast CTF4 protein consists of 927 amino acids [1], whereas the human WDHD1 protein comprises 1129 amino acids. WDHD1 contains several functional domains, including an N terminus WD40 repeat domain, a SepB domain, and a C terminus HMG-box (Figure 2A,B). It is important to highlight that the C terminus HMG-box is absent in the yeast CTF4 counterpart.

The WD40 repeat domains adopt circular β-propeller structures, which serve as pivotal mediators of protein–protein interactions [28]. The structure of WD40 repeat domains is based on a seven-fold repeat of a four-stranded antiparallel β-sheet motif. Each repeat contains approximately 40 amino acid residues and is radially arranged around a central tunnel like a propeller [29]. WD40 repeat domains frequently constitute essential subunits within multiprotein complexes involved in numerous cellular processes, including cellular growth and division, DNA damage repair, the ubiquitin–proteasome system, and chromatin organization [30]. Remarkably, WDHD1 has been demonstrated to play a crucial role in safeguarding replication forks and inducing cell death linked to double-strand breaks (DSBs), mainly through its WD40 repeat domain. Overall, the WD40 repeat domain within WDHD1 is essential for cellular proliferation and serves as a critical safeguard against genome instability.

The SepB domain corresponds to the highly conserved section in WDHD1 [31]. The SepB domain consists of a β-propeller with six blades and a bundle at the C terminus that includes five α-helices. The SepB domain promotes the trimerization of WDHD1 [32] through its β-propellers. Furthermore, the SepB domain contains the binding sites for polymerase α [11], solidifying its crucial function as the pivotal domain linking the Cdc45/Mcm2-7/GINS (CMG) complex to polymerase α.

The mammal WDHD1 protein is characterized by the presence of a distinct HMG box domain at its C terminus, which is known to participate in DNA binding mechanisms [6]. Structurally, the HMG domain of WDHD1 exhibits similarities to HMGB1, a canonical member of the HMGB subfamily known for its affinity for DNA binding [33]. Recent discoveries indicate that the HMG box of WDHD1 plays a role in positioning DNA polymerase α on the lagging strand by interacting with it [32]. Furthermore, domain analysis of WDHD1 has demonstrated the significance of the HMG box in facilitating efficient replication. Moreover, studies have confirmed the DNA binding capacity of WDHD1, primarily facilitated by its HMG domain located at the C terminus [4].

The crystal arrangement reveals that WDHD1 adopts a trimeric form (Figure 2C), with the SepB domain playing a crucial role in facilitating trimerization and serving as the main region responsible for trimer assembly [4]. WDHD1/CTF4 also exists as a trimer in solution demonstrated by multi-angle laser scattering (MALS) and native mass spectrometry [34]. Furthermore, the trimeric structure allows for flexible orientations of both the WD40 repeat domain and the HMG box.

## 4. Biological Functions of WDHD1

### 4.1. DNA Replication

Accurate and precise duplication of chromosomal DNA, which is essential for successful cellular division, is imperative for sustaining cellular life [35]. The core machinery of DNA replication is the replisome, which is composed of the replisome progression complex (RPC) and DNA polymerases [36,37,38]. RPC consists of the Cdc45/Mcm2-7/GINS complex and accessory factors, such as the Tof1/Csm3/Mrc1 complex, the histone chaperone FACT, and WDHD1 [16]. These accessory factors help effective DNA replication by removing nucleosomes before the replication forks, resolving DNA topology, and assisting in bypassing DNA damage. Essential components of the CMG complex are cell division cycle protein 45 (Cdc45), MCM helicase (MCM2-7), and Go Ichi Ni San (GINS). Working synergistically, these elements ensure efficient DNA replication by facilitating nucleosome removal ahead of the replication forks, resolving DNA topological challenges, and bypassing DNA damage [39,40].

DNA replication begins with two separate stages: the assembly of the pre-replication complex (pre-RC) and its subsequent activation [41]. Before the S phase, replication origins are licensed through the sequential binding of various elements associated with the eukaryotic DNA helicase responsible for replication. Sequential enlistment to the DNA replication origins contributes to the formation of the pre-RC complex, preparing for the subsequent initiation of DNA replication [42].

WDHD1 actively participates in the complex process of DNA replication through its dynamic interactions with DNA polymerase α and the CMG helicase complex [26,34,43]. Notably, recent findings in budding yeast have revealed the trimeric nature of WDHD1/CTF4 [44], which forms functional associations with several replication factors including Timeless, Tipin, Claspin, and MCM10 within the replication fork [3,45,46]. Moreover, WDHD1 is closely associated with important post-transcriptional processes and the formation of the pre-RC, both of which play significant roles in the initiation of DNA replication [11].

The trimeric SepB domain of WDHD1 assumes a nearly perpendicular binding orientation with the N-tier face of CMG, forming a large interface with Cdc45 and GINS components (Figure 3A). Comparison between the isolated crystal structure and the SepB domain within the core human replisome reveals insignificant differences in structure [47]. Electron microscopy analysis reveals that the WDHD1 trimer assumes a disk-shaped configuration, seamlessly interfacing with the leading face of the CMG complex in an edge-on manner and exhibiting a nearly perpendicular orientation. Despite WDHD1 being full-length, only the SepB trimerization domain [4,32] of WDHD1 is located between the CMG complex and polymerase α in this complex arrangement.

### 4.2. Cell Cycle

The process of cell division involves a series of cellular activities that result in the production of two daughter cells. In organisms with a nucleus, cell division is traditionally divided into distinct stages: G1, S, G2, and M. Crucial for regulating the progression of the cell cycle are cyclin-dependent kinases (CDKs), which are kinases that rely on cyclin subunits to regulate their activity [48]. Additionally, various essential cellular processes, including DNA duplication, genetic transcription, protein production, and post-translational modifications, exhibit a coordinated progression that aligns with the stages of the cell cycle [49].

Extensive research has confirmed the association between WDHD1 and the intricate control of cellular division. Studies have demonstrated that WDHD1 plays a role in regulating entry into the S phase, as depletion of WDHD1 leads to G1 cell cycle arrest and hinders timely progression to the S phase [10,26]. In the context of tumor cells, WDHD1 assumes a crucial role in facilitating DNA replication, promoting efficient S-phase progression, and maintaining effective mechanisms for homologous recombination repair. Depletion of WDHD1 leads to increased accumulation of DNA damage, resulting in significant delays in the S-phase progression. Consequently, the cell cycle stalls at the junction between the late S phase and the G2 phase, ultimately leading to the death of cancer cells [10]. Furthermore, research conducted by Sato et al. has demonstrated that genetic elimination of WDHD1 disrupts the initiation and progression of the S phase, ultimately leading to the death of cancer cells following cell division [19].

Hypoxia-inducible factor 1-alpha (Hif-1α) has been shown to arrest the cell cycle by blocking Myc function and increasing the levels of p21 and p27, proteins that inhibit CDK activity [50]. WDHD1 plays a crucial role in the Hif-1α-dependent checkpoint pathway, leading to cell cycle arrest in the G1 stage. During the S and G2 phases, WDHD1 also plays important roles in multiple cellular processes, including checkpoint activation, sister chromatid cohesion, and DNA damage repair. As a result, WDHD1 is essential for preserving genome integrity. Under replication stress, WDHD1 is phosphorylated at T826 by ataxia telangiectasia and Rad3-related protein (ATR), and phosphorylated WDHD1 is critical for facilitating the interaction between claspin and checkpoint kinase 1 (Chk1). This interaction enhances the effective activation of Chk1, a key protein kinase involved in the DNA damage checkpoint pathway in the G2 phase [51].

### 4.3. DNA Damage Repair

Cells possess a diverse array of repair mechanisms that facilitate the correlation of various types of DNA damage. Mammalian cells, in particular, rely on at least five prominent repair pathways, each with complementary attributes and partially overlapping functions. The pathways include nucleotide excision repair, base excision repair, mismatch repair (MMR) [52], and double-strand break repair, involving both homologous recombination (HR) repair and nonhomologous end joining [53].

The repair of DNA double-strand breaks (DSBs) primarily occurs through the process of HR, which involves the essential tumor suppressor protein CtIP in the initial step of DSB end resection, an important component of HR repair [54,55]. CtIP not only plays a role in DNA double-stranded breaks (DSBs) repair through HR but also contributes to the activation of ATR and Chk1 kinases, which are responsible for initiating the cell cycle checkpoint [56,57]. WDHD1, upon encountering DNA damage, can recruit repair proteins such as CtIP and claspin to the affected areas, thereby promoting various repair mechanisms (Figure 3B), including HR and mismatch repair [58,59,60,61]. Furthermore, the localization of WDHD1 to DSB sites depends on mediator of DNA damage checkpoint 1 (MDC1), breast and ovarian cancer susceptibility protein 1 (BRCA1), and Ataxia Telangiectasia Mutated (ATM) [62].

The presence of WDHD1 at DNA damage sites through the MDC1-RNF8 pathway is important for providing resistance against various agents that cause DNA damage and replication stress [61]. Depletion of WDHD1 makes cells highly responsive to different chemotherapy drugs, such as PARP inhibitors, which are frequently used in clinics to treat patients with prostate, ovarian, and breast cancer.

Moreover, proteomic investigations have revealed a correlation between WDHD1 and MMR. Further research using protein–protein interaction analyses confirmed that the HMG box in WDHD1, a domain-specific to vertebrates WDHD1 and of recent evolutionary origin, facilitates interactions between WDHD1 and mismatch repair protein 2 (MSH2), a key element of the MMR pathway [63]. In the MMR pathway, WDHD1 is involved in recruiting the MutS complex to chromatin during DNA replication, preventing mismatches during fork progression.

### 4.4. Embryogenesis

In addition to its regulatory functions in DNA replication and DNA damage repair, recent research has revealed the involvement of WDHD1 in promoting mouse embryo development [18]. WDHD1 is important for post-implantation embryo development in mice, and the knockdown of WDHD1 results in embryonic lethality [64]. Moreover, there is evidence of increased WDHD1 expression as undifferentiated spermatogonia transition to differentiating spermatogonia in the male germline [65]. Additionally, WDHD1 demonstrates its ability to control follicle maturation in sheep with multiple births by influencing the cell cycle, GnRH signaling pathway, and P53 signaling pathway [66].

### 4.5. Centromeric and Sister Chromatid

During mitosis, the centromere, a chromatin region determined by epigenetics, plays a vital role in ensuring the accurate separation of chromosomes [67]. To form centromeric DNA, a chromatin must incorporate centromere protein A (CENP-A). Early in the G1 phase, CENP-A is incorporated into centromeres by Holliday junction recognition protein (HJURP) and other chromatin assembly factors [68]. In this process, WDHD1 works in conjunction with HJURP to recruit CENP-A to centromeres, as shown in Figure 3C.

WDHD1 exhibits a cell cycle-dependent association with centromeres, particularly during the mid-to-late S phase. The intricate connection between DNA replication and the establishment of sister chromatid cohesion during the S phase ensures the accurate segregation of chromosomes, which is a critical mechanism in cell division. WDHD1 plays a significant role in mediating this association [69].

Additionally, non-coding RNA has been discovered to play a crucial role in the establishment and maintenance of centromeres across various eukaryotic organisms [70]. These RNAs serve as critical determinants in shaping the chromatin structure associated with centromeres [71,72]. WDHD1 functions as an important element in this RNA-dependent mechanism, which is responsible for maintaining the integrity of centromeres and ensuring genome stability [73].

### 4.6. Histone Acetylation

Histone acetyltransferases (HATs) play an important role in regulating chromatin structure and are associated with the development of various diseases, including cancer [74]. Among these HATs, General control non-derepressible 5 (Gcn5) stands out as the first transcription enzyme linked to the control of multiple cellular processes [75]. Additionally, WDHD1 has an impressive ability to regulate the stability of Gcn5 proteins, thus influencing the acetylation process of histone H3.

Several studies have reported the formation of a complex consisting of WDHD1, histone H3, and Gcn5 [59]. Through protein–protein interactions (Figure 3D), WDHD1 enhances the stability of Gcn5, facilitating the acetylation of H3K9. Furthermore, WDHD1 disrupts the interaction between Gcn5 and Cullin4-RING E3 ubiquitin ligase (CRL4), thereby stabilizing Gcn5 by preventing its ubiquitination and subsequent degradation [58].

The involvement of Gcn5 in the regulation of cancer-related gene expression, including its role in regulating the transcription of c-Myc and E2F [76], has been suggested. Moreover, Gcn5 has been identified as a suppressor of autophagy, a lysosome-dependent cellular degradation process [77]. Therefore, it is plausible that WDHD1 maintains Gcn5 stability to support the growth and survival of cancer cells.

## 5. WDHD1 and Cancer

The WDHD1 protein is not only involved in diverse physiological processes but also the initiation and progression of cancer. Emerging evidence suggests a significant increase in WDHD1 expression across different cancer types, establishing an association between WDHD1 expression and cancer progression (Table 1).

The GEPIA database (http://gepia.cancer-pku.cn/, accessed on 17 May 2023) was used to examine WDHD1 expression in both tumor and normal tissues. Figure 4A shows increased WDHD1 expression levels in various tumor types, including breast cancer (BRCA), esophageal carcinoma (ESCA), head and neck squamous cell carcinoma (HNSC), lung adenocarcinoma (LUAD), pancreatic adenocarcinoma (PAAD), and so on. Moreover, a comprehensive analysis of data from the cBioPortal database (www.cbioportal.org/, accessed on 17 May 2023) revealed a wide range of WDHD1 modifications commonly found in various tumor types, such as lung cancer, esophageal cancer, and pancreatic cancer. These modifications encompass various molecular aberrations, including mutations, amplifications, deep deletions, and multiple concurrent alterations (Figure 4B).

Bladder cancer, BLCA. Breast cancer, BRCA. Cervical squamous cell carcinoma and endocervical adenocarcinoma, CESC. Colon adenocarcinoma, COAD. Lymphoid neoplasm diffuse large B-cell lymphoma, DLBC. Esophageal carcinoma, ESCA. Glioblastoma multiforme, GBM. Head and neck squamous cell carcinoma, HNSC. Lung adenocarcinoma, LUAD. Lung squamous cell carcinoma, LUSC. Pancreatic adenocarcinoma, PAAD. Rectum adenocarcinoma, READ. Skin cutaneous melanoma, SKCM. Stomach adenocarcinoma, STAD. Testicular germ cell tumors, TGCT. Thymoma, THYM. Uterine corpus endometrial carcinoma, UCEC. Uterine carcinosarcoma, UCS.

### 5.1. WDHD1 and High-Risk HPV

Human papillomaviruses (HPVs) are small, round-shaped DNA viruses that can cause sores on the skin and mucous membranes. The presence of high-risk HPVs has been strongly associated with the onset of cervical cancer and other malignancies. Approximately 75% of cervical cancer cases can be attributed to high-risk HPV genotypes, such as 16 and 18 [86].

As an important viral oncogene in HPV, E7 can be expressed in HPV-infected cells and induce cervical cancer. E7 facilitates the degradation of the retinoblastoma protein (pRB), leading to the release of the transcription factor E2F and the activation of genes that disrupt DNA replication and cell proliferation [87]. Additionally, cells expressing E7 showed a significant increase in WDHD1 protein expression. Inhibiting WDHD1 expression resulted in a decrease in E7-induced disruption of the G1 checkpoint and hindered excessive DNA re-replication [88]. Additionally, WDHD1 levels were approximately twice as high in E7-expressing cells compared to control cells, and its reduction led to a specific G1 arrest in E7-expressing cells. These findings provide insights into how HPV influences the cell cycle and may contribute to the development of HPV-targeted treatments.

### 5.2. WDHD1 and Nasopharyngeal Carcinoma

Nasopharyngeal carcinoma (NPC) is an aggressive and metastatic malignancy that originates from the nasopharynx region [89]. Notably, there has been a concerning increase in the age of onset, leading to significant financial burdens for patients’ families and society at large [90]. In NPC, WDHD1 expression levels are notably upregulated, suggesting its potential effect on the dynamics of the cell cycle through the regulation of integrin alpha V (ITGAV) expression [78]. Furthermore, bioinformatic analyses have indicated that WDHD1 could serve as a discriminatory biomarker for distinguishing the NPC cohort from the non-cancer control group, holding promise as a novel screening molecule for NPC.

### 5.3. WDHD1 and Laryngeal Cancer

Laryngeal cancer (LC) is a prominent cancer in the head and neck region, ranking second in terms of occurrence [90]. Current therapeutic approaches for LC include surgical interventions, radiotherapy, chemotherapy, and biological immunotherapy. Despite advancements in treatment, the survival rate for laryngeal squamous cell carcinoma (LSCC) remains poor due to its propensity for local invasion and metastasis [91]. Surprisingly, chromatin immunoprecipitation and RNA-seq methods have highlighted the remarkable ability of WDHD1 to differentiate between LSCC and non-LSCC cases. Moreover, WDHD1 protein expression significantly increases in individuals with advanced disease stages and the presence of lymph node metastasis. Additionally, WDHD1 may impact the initiation and progression of LSCC by positively regulating S-phase kinase-associated protein 2 (Skp2) [79]. Overall, WDHD1 represents a promising therapeutic target for laryngeal cancer.

### 5.4. WDHD1 and Esophageal Cancer

Esophageal cancer is a common form of cancer worldwide, known for its high occurrence, aggressive nature, significant death rates, and poor outlook. Despite the continuous progress in medical understanding and the constant enhancement of treatment approaches, the five-year survival rate for esophageal cancer has shown minimal improvement. This persistent challenge necessitates further exploration to enhance patient outcomes [92,93]. The identification of WDHD1 as a central gene linked to esophageal squamous cell carcinoma (ESCC) highlights its crucial involvement in the onset of esophageal cancer [19,81]. Significantly, WDHD1 acts as a target downstream of the PI3K/AKT pathway [80], playing a significant role in the progression of esophageal cancer by participating in abnormal cellular regulation through multiple downstream targets of AKT.

### 5.5. WDHD1 and Lung Cancer

Lung cancer is a widespread malignancy with a significant impact on global health, affecting approximately 18 million individuals each year and resulting in 16 million deaths [94,95]. WDHD1 overexpression is inversely associated with the overall survival of lung cancer patients, indicating its potential prognostic relevance [19]. Additionally, suppressing WDHD1 expression leads to increased radiosensitivity in non-small cell lung carcinoma (NSCLC) cells, suggesting its effect on the modulation of therapeutic response [25]. Gou et al. [24] further revealed the upregulation of MAPRE2 following WDHD1 knockout and identified a protein–protein interaction between WDHD1 and MAPRE2 in nuclear. WDHD1 promotes the ubiquitination of MAPRE2 and contributes to the development of cisplatin resistance. These findings highlight the potential value of targeting WDHD1 and MAPRE2 to overcome cisplatin resistance in lung cancer.

### 5.6. WDHD1 and Breast Cancer

Breast cancer is the leading cause of cancer-related deaths among women worldwide [96]. Triple-negative breast cancer (TNBC), which accounts for approximately 10–20% of breast cancer cases [97,98], lacks expression of estrogen receptor, progesterone receptor, and human epidermal growth factor receptor 2. TNBC exhibits an aggressive phenotype with high-grade characteristics, predisposing patients to an increased risk of tumor recurrence and metastasis [99]. In TNBC patient samples, WDHD1 expression levels are significantly elevated compared to normal breast tissues. Furthermore, WDHD1 expression is associated with tumor size, stage, and proliferative capacity. WDHD1 plays an important role in the survival of TNBC cells, making it a potential indicator or target for TNBC [22].

### 5.7. WDHD1 and Hepatocellular Carcinoma

Liver cancer, an increasingly prevalent malignant neoplasm, poses a significant threat to human well-being, with an annual incidence of approximately 2–3% and a poor five-year survival rate of only 18% globally [94]. The majority of liver cancer cases (75% to 85%) originate from hepatocellular carcinoma (HCC), the most common form of liver cancer [100]. HCC presents a formidable challenge due to its high occurrence and death rates, complex mechanisms, and therapeutic constraints [101,102]. Notably, WDHD1 has been found to exhibit increased protein and mRNA expression levels, which are associated with a negative prognosis [82]. The interaction between WDHD1 and UBA52 may contribute to the oncogenic role of WDHD1 in HCC initiation and progression, suggesting its involvement in tumorigenesis and malignant progression. Therefore, WDHD1 may serve as a therapeutic target for HCC.

### 5.8. WDHD1 and Cholangiocarcinoma

Cholangiocarcinoma (CCA), also known as bile duct cancer, is a malignant neoplasm originating from cholangiocytes located at different sites along the biliary tree, exhibiting diverse differentiation markers [103]. This rare yet aggressive carcinoma is commonly regarded as a challenging tumor, displaying increased incidence, mortality rates, and an unfavorable prognosis globally, particularly prevalent in East Asia [104]. Notably, reducing WDHD1 expression has been shown to hinder the process of epithelial–mesenchymal transition (EMT) and the spread of cancer cells to lymph nodes in CCA, making it a potential target for CCA treatment [20].

### 5.9. WDHD1 and Pancreatic Adenocarcinoma

Pancreatic adenocarcinoma, a major contributor to global cancer-related mortality, presents significant challenges in diagnosis and treatment [96]. The elusive nature of this disease often leads to late-stage detection, with most patients already having metastases at the time of diagnosis. The lack of reliable diagnostic biomarkers and therapeutic targets further complicates the effective screening and management of pancreatic adenocarcinoma. By applying weighted correlation network analysis (WGCNA) to expression profiles obtained from The Cancer Genome Atlas (TCGA), a comprehensive gene interaction network was established, revealing multiple genes, including WDHD1, with potential implications in pancreatic adenocarcinoma [83]. WDHD1 expression was significantly increased in pancreatic cancer and had a 10% amplification mutation frequency, suggesting its role as an oncogene. In addition, the high expression of WDHD1 can promote DNA replication and DNA repair in cancer cells, thus inducing chemotherapy resistance of pancreatic cancer. Survival analysis highlighted the significant impact of WDHD1 on the diagnosis and treatment of this disease, underscoring its clinical relevance. In summary, WDHD1 is an important potential diagnostic marker and therapeutic target for pancreatic cancer.

### 5.10. WDHD1 and Cervical Cancer

Cervical cancer, the fourth most common cancer in women worldwide, is associated with significant rates of illness and mortality [105]. Persistent human papillomavirus (HPV) infections are primarily responsible for the development of cervical tumors. The expression of viral oncoproteins, including E6 and E7, encoded by HPV genetic material, contributes to epigenetic dysregulation, influencing the initiation and spread of cervical cancer [106]. WDHD1 is highly expressed in cervical cancer cells and plays an important role in the initiation of cervical cancer induced by HPV E7. Remarkably, Chen et al. identified the clinical significance and potential prognostic value of WDHD1, an important gene, particularly in relation to lymph node metastasis in cervical cancer [21]. WDHD1 is a potential therapeutic target in cervical cancer progression and metastasis.

### 5.11. WDHD1 and Ovarian Cancer

Ovarian cancer, a prevalent gynecological malignancy, poses a significant challenge due to its asymptomatic nature, leading to late-stage detection and poor prognosis [90]. The five-year survival rate for advanced ovarian cancers is 30%, with most patients succumbing to the disease within two years of diagnosis [107]. Cisplatin, a chemotherapeutic agent, exerts cytotoxic effects by inducing DNA cross-links, inhibiting DNA replication and transcription processes [108]. WDHD1 can activate fanconi anemia (FA) signaling in ovarian cancer and leads to cisplatin resistance [84]. Additionally, phosphorylation of WDHD1 by the ATR is important for enhancing therapeutic outcomes in platinum-resistant ovarian cancer [23]. Further investigation is warranted to explore the potential of inhibiting WDHD1 as a promising therapeutic approach to overcome platinum drug resistance in ovarian cancer.

### 5.12. WDHD1 and Acute Myeloid Leukemia

Acute myeloid leukemia (AML) exhibits significant genetic heterogeneity, presenting challenges in the search for novel therapeutic approaches [109]. Inhibition of WDHD1 has shown significant effects on the proliferation and survival of primary leukemic cells, highlighting its potential as a promising target for AML treatment [85]. Additional investigation is necessary to determine the potential of WDHD1 as a therapeutic approach for AML.

## 6. Molecular Mechanisms of Action of WDHD1

### 6.1. PI3K/AKT Pathway

The PI3K/AKT pathway, known for its crucial involvement in cell survival, proliferation, autophagy, and apoptosis [110], has been extensively investigated with regard to its role in the regulation of normal cellular processes and malignant tumors [111]. Dysregulation of the PI3K/AKT signaling results in abnormal signal transduction, thereby triggering the development of several diseases, including diabetes, cardiovascular disorders, neurological ailments, and hematological conditions [112]. Conversely, activation of the PI3K/AKT pathway is recognized as a prominent hallmark of cancer [113].

The study by Sato et al. identified a conserved phosphorylation site in the WDHD1 protein, which influences the AKT kinase activity [19]. In the study, they performed experiments involving immunoprecipitation of labeled WDHD1 and recombinant AKT1 protein as a kinase. Results showed that AKT1 directly phosphorylated WDHD1 as determined by the kinase analysis using PAS antibody immunoblotting [80]. These findings present convincing evidence that WDHD1 functions as a downstream target of the PI3K/AKT pathway (Figure 5).

### 6.2. JAK-STAT Signaling Pathway

A group of transcription factors known as signal transducers and activators of transcription (STAT) have been implicated in the regulation of various cellular processes and influence the development of multiple human diseases [114]. Among the STAT protein family, STAT3 stands out as a vital member known to modulate proliferation, differentiation, survival, immunosuppression, angiogenesis, and tumorigenesis [115,116].

Recent studies found that upregulation of STAT3 and WDHD1 in cells increased the expression of oncogenes [88]. The STAT protein family exhibits a preference for a common DNA binding pattern characterized by TTCC (C or G) GGAA (or typically TTN5AA) [117]. Notably, the WDHD1 promoter region contains three potential binding sites for STAT3. It was previously reported that STAT3 can attach to each of the three presumed binding locations within the WDHD1 promoter area in MCF-7 cells. This demonstrates a link between STAT3 and the promoter region of WDHD1, confirming that the *WDHD1* gene is targeted by STAT3 (Figure 5), which regulates the DNA replication function of STAT3.

## 7. Inhibitors of WDHD1

Genetic analyses conducted in yeast and CRISPR/Cas9 screening performed in human cells have yielded compelling evidence that WDHD1 is a promising target gene for cancer therapeutics [118,119]. The median dependency score of WDHD1 shown in the DepMap database (https://depmap.org/portal/, accessed on 13 May 2023) is lower than −1, revealing the essential role of WDHD1 in the growth and proliferation of pan-cancer cells. There is evidence that WDHD1 assumes a critical role in the repair of DSBs by governing HR repair mechanisms [17,61]. Consequently, inhibition of WDHD1 may be an effective strategy for augmenting the sensitivity of treatment modalities that induce DSBs, such as ionizing radiation. It can also guide the selection of chemotherapy agents, thereby enhancing their ability to eliminate cancer cells.

Li et al. utilized a high screening platform and a luciferase reporter assay that targeted WDHD1, while also making use of the Library of Pharmacologically Active Compounds. Bazedoxifene acetate (BZA) and an uncharacterized compound [(E)-5-(3,4-dichlorostyryl)benzo[c][1,2]oxaborol-1(3H)-ol] (CH3, Figure 6A,B) [23] were found to be the two potent inhibitors of WDHD1. These compounds inhibit WDHD1 by promoting its degradation. Surprisingly, CH3 hinders the polymerization process of WDHD1 by directly interacting with its WD40 repeat domain (Figure 6C). Depolymerization improves the connection between WDHD1 and the E3 ligase Cullin 4B, thereby promoting the ubiquitination and subsequent breakdown of WDHD1.

Moreover, CH3 inhibits the growth of various cancer types. For example, inhibitors of WDHD1 can enhance the responsiveness of platinum-resistant ovarian cancer cells to platinum-based medications. Furthermore, the CH-3 compound, a WDHD1 inhibitor, causes cell cycle arrest at the G2/M stage by altering the ATM signaling pathway. It also amplifies radiation-induced DNA damage by inhibiting the DNA HR repair pathway [23]. Moreover, in vivo studies showed that CH-3 boosts the sensitivity of NSCLC cells to radiation [25], indicating that radiosensitizers targeting WDHD1 may provide an alternative approach to inhibit NSCLC progression. Altogether, these findings suggest that utilization of WDHD1 inhibitors may be effective comprehensive anti-cancer treatments.

## 8. Future Perspectives

In this comprehensive review, we provide a detailed overview of the recent advancements in terms of the biological functions and molecular mechanisms associated with WDHD1 (Table 2). Extensive research has shed light on the pivotal involvement of WDHD1 in vital cellular processes, encompassing DNA replication initiation, post-transcriptional events governing centromere formation, DNA damage repair, and cell cycle progression. Significantly, recent investigations have unveiled compelling evidence establishing a clear association between WDHD1 and the pathogenesis as well as progression of multiple cancer types. Consequently, these findings offer promising avenues for harnessing the diagnostic and therapeutic potential of WDHD1 in the clinical setting.

For example, Li et al. discovered that CH3, an inhibitor of WDHD1, restores platinum-sensitivity to platinum-resistant ovarian cancer cells [23]. Similarly, Gong and Xiao provided evidence suggesting that cisplatin resistance can be induced by WDHD1 through ubiquitinating MAPRE2 [24]. Moreover, Gou et al. observed a radiosensitivity enhancement in NSCLC cells after exposure to CH-3 in vivo [25]. Overall, these investigations substantially expand our understanding of the clinical value of WDHD1.

Although extensive investigations have revealed the function and mechanism of WDHD1, further research is needed to explore its precise molecular regulatory mechanisms and clinical application. For instance, although some studies have led to the development of WDHD1 inhibitors, further investigations are required to determine the feasibility of such inhibitors in clinical settings. In addition, current research has primarily focused on the physiological functions of WDHD1 in tumorigenesis, with limited attention to its underlying mechanisms and potential implications in non-tumor diseases. Next, although some proteins interacting with WDHD1 have been identified, novel protein interactions and their associated functions remain to be explored. Lastly, few studies have explored the signaling pathways modulated by WDHD1, and its potential involvement in other signaling pathways needs to be investigated. In summary, WDHD1 is a valuable indicator and target for the treatment of different types of cancers, and comprehensive research into its functional and mechanistic aspects is advocated.

## Figures and Tables

**Figure 1 ijms-24-12494-f001:**
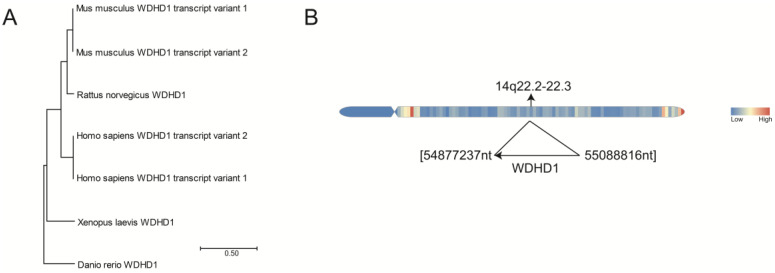
The phylogenetic tree analysis of WDHD1 in different species and the location of WDHD1. (**A**) Phylogenetic tree analysis of WDHD1 in Mus musculus, Rattus norvegicus, Homo sapiens, Xenopus and Danio. (**B**) WDHD1 is located on chromosome 14q22.2–2.3. Areas highlighted in red signify high gene density.

**Figure 2 ijms-24-12494-f002:**
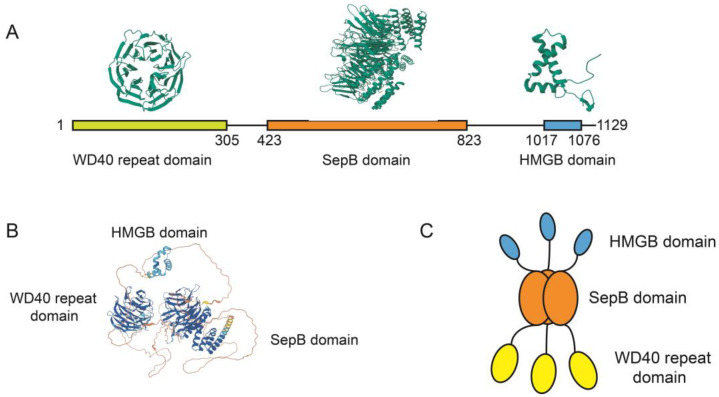
The structure of WDHD1. (**A**) The structure of WD40 repeat domain (PDB code 5GVA), HMG-box domain (PDB code 2D7L), and SepB domain (PDB code 5GVB) in WDHD1 are obtained from the PDB database (https://www.rcsb.org/, accessed on 2 May 2023). (**B**) The structure of WDHD1 (AF-O75717-F1) is obtained from the AlphaFold database (https://alphafold.ebi.ac.uk/, accessed on 2 May 2023). (**C**) The WDHD1 forms trimer in cells.

**Figure 3 ijms-24-12494-f003:**
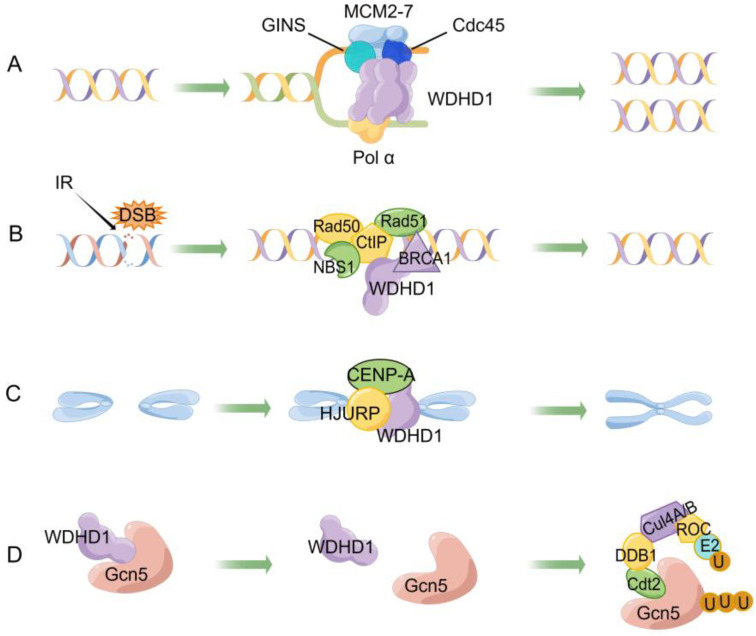
The biological functions of WDHD1. This image is drawn by Figdraw. (**A**) WDHD1 participates in DNA replication by forming a replisome with the Cdc45/Mcm2-7/GINS complex and polymerase α. When a replisome is present, one parental DNA forms two progeny DNA. (**B**) Ionizing radiation (IR) leads to double-strand DNA breaks (DSBs). WDHD1 binds to CtIP, recruiting Rad50, Rad51, NSB1, BRCA1, and participates in DNA homologous repair to repair DSBs. (**C**) In the metaphase of mitosis, WDHD1 works in conjunction with HJURP to recruit CENP-A to format centromeres, making two sister chromatids connect together. (**D**) Combining WDHD1 with Gcn5 can avoid the ubiquitination degradation of Gcn5. When WDHD1 and Gcn5 are isolated, Gcn5 will be ubiquitylated and degraded by the Cdt2-DDB1-Cul4A/B-Roc complex. E2 represents a ubiquitin-conjugating enzyme, and U represents ubiquitin.

**Figure 4 ijms-24-12494-f004:**
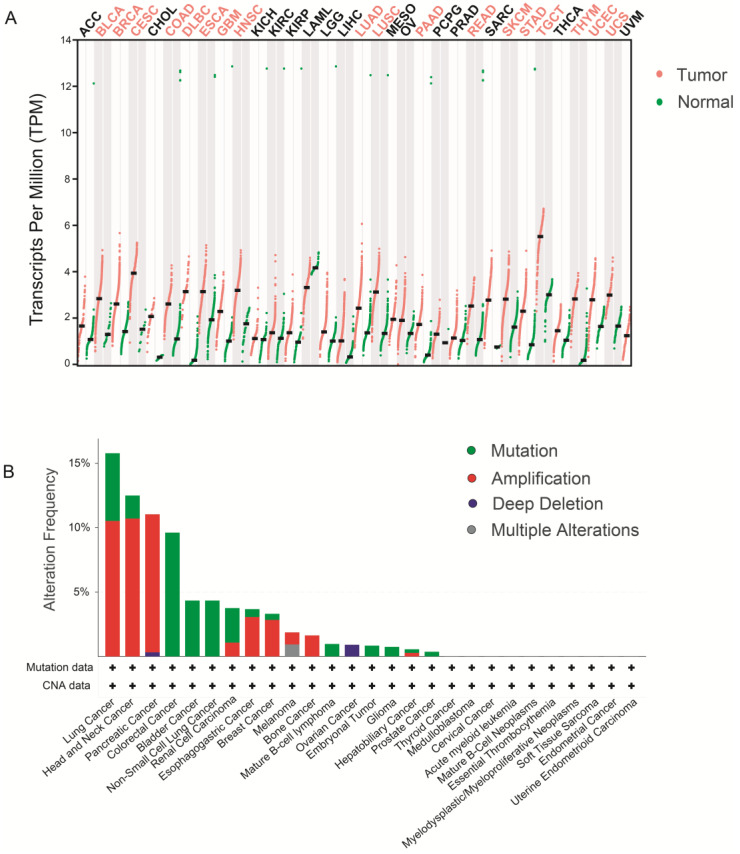
The expression and mutation frequency of WDHD1 in tumors. (**A**) The differential expression levels of WDHD1 in tumor and normal tissues. Gene expression levels were determined by TPM value, and the cancer type is shown above. Red dots represent tumor samples and green dots represent normal samples. WDHD1 has high expression in most tumors, including BLCA, BRCA, CESC, COAD, DLBC, ESCA, GBM, HNSC, LUAD, LUSC, PAAD, READ, SKCM, STAD, TGCT, THYM, UCEC, UCS. The cancers with *p* < 0.05 are shown in red font. (**B**) WDHD1 alteration frequency, including mutation data and copy number alteration (CNV) data in various tumors. The cancer type is shown below. Green, mutation frequency; red, amplification frequency; blue, deep deletion frequency; grey, multiple alterations frequency.

**Figure 5 ijms-24-12494-f005:**
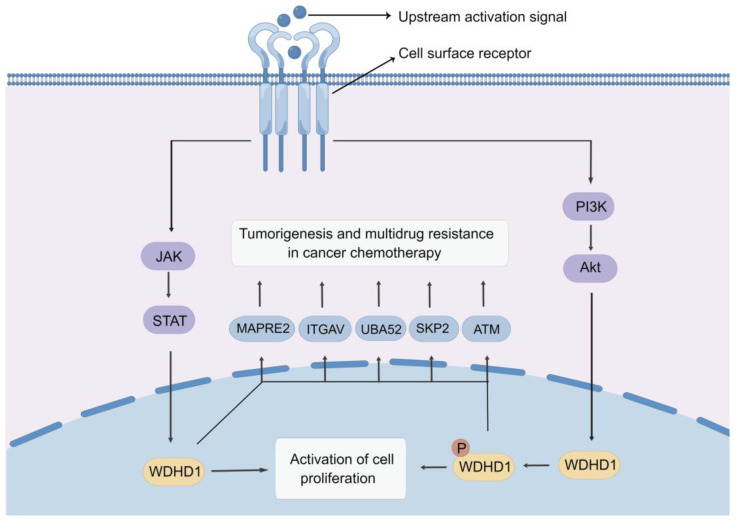
Molecular mechanisms of action of WDHD1, drawn by Figdraw.

**Figure 6 ijms-24-12494-f006:**
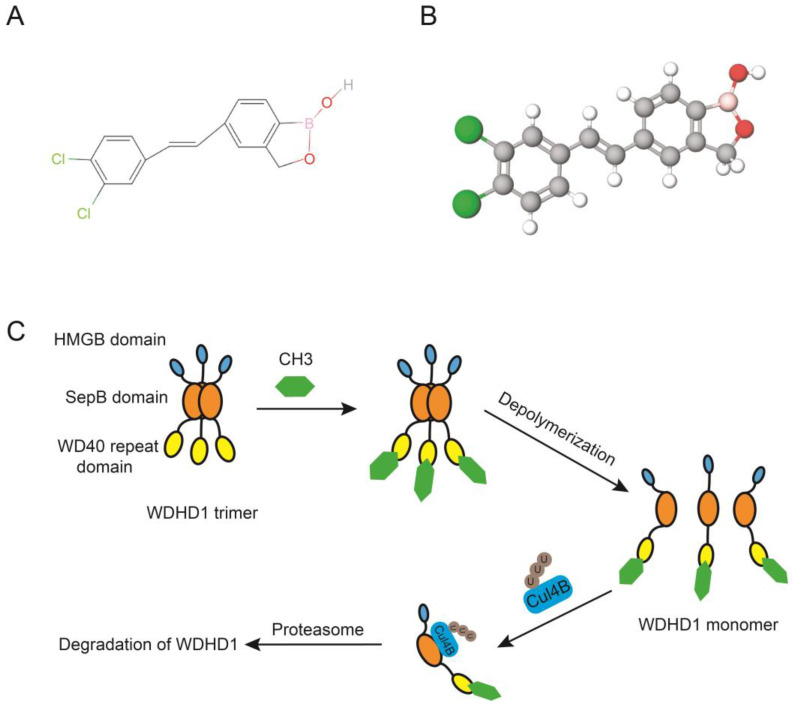
Chemical structure of compound “CH3” and the inhibitory mechanism. (**A**) The chemical structure of compound “CH3”. (**B**) The 3D chemical structure of CH3. The green ball represents Cl atoms. The grey ball represents C atoms. The white ball represents H atoms. The red ball represents O atoms. The pink ball represents B atoms. (**C**) CH3 inhibit WDHD1 by promoting its degradation. CH3 directly interacts with the WD40 repeat domain of WDHD1 and leads to the depolymerization of WDHD1 trimers to monomers, which improves the connection between WDHD1 and the E3 ligase Cullin 4B, promoting the ubiquitination and subsequent degradation of WDHD1.

**Table 1 ijms-24-12494-t001:** WDHD1 is associated with different cancers.

Tumor	Mechanisms	Functions	Refs.
Nasopharyngeal carcinoma	Regulating ITGAV expression	Affect cell apoptosis	[78]
Laryngeal squamous cell carcinoma	Regulating Skp2 expression	Tumor diagnostic biomarkers and tumor formation	[79]
Esophageal cancer	As the downstream target of the PI3K/AKT pathway	The occurrence and survival of tumors	[80]
	As a cell cycle regulator and a downstream molecule in the PI3K/AKT pathway	The occurrence and survival of tumors	[19]
	Weighted gene co-expression network analysis	As a hub gene	[81]
Lung adenocarcinoma	Enhancing the ubiquitination degradation of MAPRE2	Cisplatin resistance	[24]
	As a cell cycle regulator and a downstream molecule in the PI3K/AKT pathway	The occurrence and survival of tumors	[19]
	Regulating the ATM signaling pathway	Radiosensitivity of cancer	[25]
Triple-negative breast cancer	As a downstream target of PTEN-AKT signaling	Tumor cells survival	[22]
Hepatocellular carcinoma	Co-expressed genes analysis	As a biomarker for diagnosis and prognosis	[82]
Cholangiocarcinoma	Inhibited by miR-494	Tumor formation and metastasis	[20]
Pancreatic adenocarcinoma	Weighted gene co-expression network analysis	As a hub gene for tumor progression	[83]
Cervical cancer	Weighted gene co-expression network analysis	As a hub gene for cancer lymph node metastasis	[21]
Ovarian cancer	Regulated by ATM- and Rad3-related pathway	Cisplatin resistance	[23]
	Activating the FA pathway	Cisplatin resistance	[84]
Acute myeloid leukemia	A large-scale loss of-function RNA interference	Tumor growth and viability	[85]

Integrin alpha V, ITGAV; S-phase kinase-associated protein 2, Skp2; Microtubule-associated protein RP/EB family member 2, MAPRE2; Ataxia-telangiectasia mutated, ATM; Fanconi anemia, FA.

**Table 2 ijms-24-12494-t002:** The upstream and downstream factors of WDHD1.

Upstream or Downstream	Related Factors	Function and Mechanism	Refs.
Upstream	miR-494	miR-494 overexpression suppresses EMT, tumor formation, and LNM while promoting CCA cell apoptosis through inhibiting WDHD1 in CCA.	[20]
	JAK-STAT pathway	WDHD1 is a novel STAT3 target gene and mediated the DNA replication function of STAT3.	[120]
	PI3K/AKT pathway	As a downstream molecule in the PI3K/AKT pathway, WDHD1 can be phosphorylated by AKT1.	[19,22,80]
Downstream	MAPRE2	WDHD1 induces cisplatin resistance in LUAD by promoting MAPRE2 ubiquitination.	[24]
	ITGAV	ITGAV may act as a potential target gene for WDHD1 and affect cell cycle in nasopharyngeal carcinoma.	[78]
	UBA52	WDHD1 may regulate UBA52 and contribute to the progress of HCC.	[82]
	SKP2	WDHD1 may affect cell cycle in LSCC by regulating SKP2.	[79]
	ATM	WDHD1 activates the ATM-CHK1-CDC25C pathway and regulates the G2/M phase block.	[23]

## Data Availability

Data are contained within the article.

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
