# Peer review of "WD Repeat and HMG Box DNA Binding Protein 1: An Oncoprotein at the Hub of Tumorigenesis and a Novel Therapeutic Target"

_ijms, 2023, doi:10.3390/ijms241512494_

Round 1

Reviewer 1 Report

This manuscript provides an overview of the overall structure, biological functions, and regulatory mechanisms of the WD repeat and HMG-box DNA binding protein 1(WDHD1). This manuscript also adresses the relevance of WDHD1 to cancer. Considering the importance and novelty of WDHD1 with respect to cancer therapies, this manuscript significantly contributes to the field of cancer treatment. I suggest several things to improve the quality of this manuscript.

1. In Abstract, it seems somewhat inappropriate that the initialism 'WDMD1' comes first in the sentence. Rephrasing it would be better.

2. Scientific names should be written in italic type (page 1, line 29).

3. If CTF4 is an initialism, it should be located after the full-length word (page 1, line 30).

4. 'N-terminal' and 'C-terminal' should be fixed to 'N terminus' and 'C terminus', respectively (page 1, lines 34 and 35).

5. Is Section 2 (Evolutionary consevation of WDHD1) the authors' analysis? If so, please state it with appropriate words. If not, please cite references.

6. The authors state that WDHD1 adopts a trimeric form, based on the crystal environment (page 3, line 107). Are there any biochemical results to prove its trimeric state in solution? If there are, please add them to the manuscript.

7. In Section 6, schematic diagrams to illustrate the molecular mechanisms of action would help readers understand them more easily.

8. The chemical structure of compound 'CH3' along with a schematic diagram to illustrate its inhibitory mechanism can be added to Section 7.

9. The word 'et al.' should be written in italic type.

Minor editing of English language is required

Author Response

We would like to express our sincere thanks to the reviewers for the constructive and positive comments.

Comment 1: In Abstract, it seems somewhat inappropriate that the initialism 'WDMD1' comes first in the sentence. Rephrasing it would be better.

Our Reply: We agree with this suggestion. We have now amended this sentence to “WD repeat and HMG-box DNA binding protein 1 (WDHD1) is a highly conserved gene from yeast to humans”. Page 1, line 12.

Comment 2: Scientific names should be written in italic type (page 1, line 29).

Our Reply: We thank the reviewer for this suggestion. We have adjusted the font on page 1, line 29.

Comment 3: If CTF4 is an initialism, it should be located after the full-length word (page 1, line 30).

Our Reply: We thank the reviewer for this suggestion. The sentence now reads: “In the model organism Saccharomyces cerevisiae (budding yeast), the human counterpart of WDHD1 is referred to as chromosome transmission fidelity factor 4 (CTF4).

Comment 4: 'N-terminal' and 'C-terminal' should be fixed to 'N terminus' and 'C terminus', respectively (page 1, lines 34 and 35).

Our Reply: We appreciate and agree with this suggestion. We have made adjustments on page 1, lines 34 and 35, and in the following text.

Comment 5: Is Section 2 (Evolutionary consevation of WDHD1) the authors' analysis? If so, please state it with appropriate words. If not, please cite references.

Our Reply: We thank the reviewer for this suggestion. We have added corresponding references and descriptions in Section 2. Page 2, line 61 and line 64.

Comment 6: The authors state that WDHD1 adopts a trimeric form, based on the crystal environment (page 3, line 107). Are there any biochemical results to prove its trimeric state in solution? If there are, please add them to the manuscript.

Our Reply: We thank the reviewer for this suggestion. WDHD1/CTF4 also exists as a trimer in solution demonstrated by multi-angle laser scattering (MALS) and native mass spectrometry. We have included the information on page 3, lines 113-115.

Comment 7: In Section 6, schematic diagrams to illustrate the molecular mechanisms of action would help readers understand them more easily.

Our Reply: We agree with this comment and thank the reviewer for this suggestion. We have added Figure 5 on page 15 as schematic diagrams of section 6.

Comment 8: The chemical structure of compound 'CH3' along with a schematic diagram to illustrate its inhibitory mechanism can be added to Section 7.

Our Reply: We thank the reviewer for this valuable suggestion. We have added Figure 6 on page 16 to show the chemical structure of compound 'CH3' along with a schematic diagram to illustrate its inhibitory mechanism.

Comment 9: The word 'et al.' should be written in italic type.

Our Reply: We thank the reviewer for this valuable suggestion. We have changed the word 'et al.' to 'et al.' in the article.

We thank the anonymous reviewers again for their constructive comments. All modifications have been marked in red.

Reviewer 2 Report

The article «WD Repeat and HMG Box DNA Binding Protein 1: An Onco-protein at the Hub of Tumorigenesis and a Novel Therapeutic Target» by Zhiwei Zhang and Qing Zhu is done at a good level and may be published in the «International Journal of Molecular Sciences». However, there are some comments I have to make.

MAJOR REVISION

1. Title of article, Lines 26-27

«WD Repeat and HMG Box DNA Binding Protein 1: An Onco-protein at the Hub of Tumorigenesis and a Novel Therapeutic Target»

«The WD repeat and HMG-box DNA binding protein 1 (WDHD1), also known as acidic  nucleoplasmic DNA-binding protein 1 (AND-1)»

A more correct characteristic of the WDHD1 protein is "a protein containing DNA-binding HMG Box Domain". The expression "HMG-Box DNA Binding Protein 1" describes the HMGB1 protein. Corrections should be made in the title and in the text of the article.

2. Lines 38-39

«The HMG domains are DNA-binding motifs commonly found in nonhistone proteins [6]».

DNA-binding HMG domains are found only in the HMGB protein group. In other nonhistone proteins, they are not present. Please clarify in the text.

3. Lines 77-78

«The human WDHD1 protein is considered the homologous gene of the budding yeast CTF4.»

Can protein be considered a gene? Please clarify in the text.

4. Please explain the concept «a β-propeller» (line 93), «CMG complex» (line 97), «WDR domains» (line 84), «WD40 domain» (lines 80, 89) («WD-40 domain», line 35) и «WD repeat» (line 26) in the text. Describe the structural features of WD domain. Please introduce uniform designations.

5. Line 95

«the SepB domain contains a positively charged groove»

What do the authors mean by "a positively charged groove"?

6. Lines 122-123

«The replisome progression complex, comprising the CMG complex and accessory factors, constitutes the core machinery for DNA replication.»

What additional factors are involved in the replisome progression complex? What is this complex? Please clarify in the text.

7. You must enter an explanation of the abbreviations: MDC1, BRCA1, ATM, ATR, ITGAV, TNBC, WGCNA etc.

8. Lines 206-208

«Further research using protein-protein interaction analyses confirmed that the HMG box, a domain-specific to vertebrates and of recent evolutionary origin, facilitates interactions between WDHD1 and mismatch repair protein 2 (MSH2)»

HMGB-domain proteins are found in all eukaryotes and it is incorrect to say that "the HMG box, a domain-specific to vertebrates and of recent evolutionary origin". If the authors want to describe evolutionary differences in organisms, then it is worth introducing additional information on this topic.

9.  Lines 213-217

«In addition to its regulatory functions in DNA replication and DNA damage repair, recent research has revealed the involvement of WDHD1 in promoting mouse embryo development [18]. WDHD1 is important for post-implantation embryo development in mice. Furthermore, the knockdown of WDHD1 during larval development results in lethality [62]»

Here the authors write about post-implantation embryo development in mice. The concept of "larval development" is given incorrectly.

10. GCN or Gcn, CDC45 or Cdc45?

11. Lines 273-274

respectively.

12. Lines 299

Please clarify in the text: «specific HPV genotypes, such as 16 and 18», E7 and HPV E7.

13. Line 351

Please clarify in the text: «a nuclear interaction between WDHD1 and MAPRE2»

14. Lines 395-396, 403-405

It is necessary to describe in more detail the effect of WDHD1 on the diagnosis and treatment of pancreatic adenocarcinoma disease and cervical cancer, underscoring its clinical relevance in the text.

15. Line 458-460

 «Analyses of dependency scores have revealed that WDHD1 has a negative dependency score in different types of cancer, highlighting its essential role in the growth and proliferation of pan-cancer cells»

It is necessary to clarify in the text: «WDHD1 has a negative dependency score in different types of cancer»

16. Figure 2

The legend should be more detailed: the information in the legends to the Figure is not enough to understand of the presented material.

HMG (panel C) and HMG box (panel B) must be replaced by a HMGB domain. Panel C: SepB and WD40 must be replaced by a SepB domain and WD40 domain respectively.

17. Figure 4

Panel A - The labels on the axis X are not visible.

The information in the legends to the Figure is not enough to understand of the presented material.

18. Figure 3

The information in the legends to the Figure is not enough to understand of the presented material.

19. References

№ 30. «doi:Doi» must be replaced by «doi»

Author Response

We would like to express our sincere thanks to the reviewers for the constructive and positive comments.

Comment 1: Title of article, Lines 26-27

«WD Repeat and HMG Box DNA Binding Protein 1: An Onco-protein at the Hub of Tumorigenesis and a Novel Therapeutic Target»

«The WD repeat and HMG-box DNA binding protein 1 (WDHD1), also known as acidic  nucleoplasmic DNA-binding protein 1 (AND-1)»

A more correct characteristic of the WDHD1 protein is "a protein containing DNA-binding HMG Box Domain". The expression "HMG-Box DNA Binding Protein 1" describes the HMGB1 protein. Corrections should be made in the title and in the text of the article.

Our Reply: We agree with this suggestion. However, the full name of WDHD1 is found to be “WD repeat and HMG-box DNA binding protein 1” in the NCBI database (https://www.ncbi.nlm.nih.gov/gene/11169) and other articles. Although this full name may be confused with HMGB1, we think “WD Repeat and HMG Box DNA Binding Protein 1” is understandable. We sincerely thank the reviewer for the comment.

Comment 2: Lines 38-39

«The HMG domains are DNA-binding motifs commonly found in nonhistone proteins [6]».

DNA-binding HMG domains are found only in the HMGB protein group. In other nonhistone proteins, they are not present. Please clarify in the text.

Our Reply: We thank the reviewer for this suggestion. The sentence now reads: “The HMG domains are DNA-binding motifs found in the HMGB protein group.” Page 1, line 39.

Comment 3: Lines 77-78

«The human WDHD1 protein is considered the homologous gene of the budding yeast CTF4.»

Can protein be considered a gene? Please clarify in the text.

Our Reply: We thank the reviewer for this suggestion. The sentence now reads: “The human WDHD1 is considered the homologous gene of the budding yeast CTF4.” Page 3, line 78.

Comment 4: Please explain the concept «a β-propeller» (line 93), «CMG complex» (line 97), «WDR domains» (line 84), «WD40 domain» (lines 80, 89) («WD-40 domain», line 35) и «WD repeat» (line 26) in the text. Describe the structural features of WD domain. Please introduce uniform designations.

Our Reply: We agree with this suggestion.

β-propeller: The β-propeller fold has been found in and is predicted to be in many different structures. The structure of the WD40 repeat domain is based on a 7-fold repeat of a four-stranded antiparallel β-sheet motif. The twisted β sheets are radially arranged around a central tunnel and they pack face-to-face like a propeller.

CMG complex: We have replaced 'CMG complex' with 'Cdc45/Mcm2-7/GINS complex'. It is a complex formed by Cdc45/Mcm2-7/GINS in the process of DNA replication.

WDR domains: We have replaced 'WDR domains' with 'WD repeat domains'.

WD40 domain: The structure of WD40 repeat domains is based on a 7-fold repeat of a four-stranded antiparallel β-sheet motif. Each repeat contains approximately 40 amino acid residues and is radially arranged around a central tunnel like a propeller.

WD repeat: The WD40 repeat domain is based on a 7-fold repeat of 40 amino acid residues in a single repeat.

We have made corresponding modifications on page 3, lines 85-88.

Comment 5: Line 95

«the SepB domain contains a positively charged groove»

What do the authors mean by "a positively charged groove"?

Our Reply: We thank the reviewer for this valuable comment. The binding site of WDHD1 and polymer α was found to be a positively charged groove on the spatial structure of the SepB domain. "a positively charged groove" is a description of the binding site. Page 3, line 99.

Comment 6: Lines 122-123

«The replisome progression complex, comprising the CMG complex and accessory factors, constitutes the core machinery for DNA replication.»

What additional factors are involved in the replisome progression complex? What is this complex? Please clarify in the text.

Our Reply: We thank the reviewer for this valuable comment. The core machinery of DNA replication is the replisome, which is composed of the replisome progression complex (RPC) and DNA polymerases. RPC consists of the Cdc45/Mcm2-7/GINS complex and accessory factors, such as Tof1/Csm3/Mrc1 complex, the histone chaperone FACT, and WDHD1. These accessory factors help effective DNA replication by removing nucleosomes before the replication forks, resolving DNA topology, and assisting in bypassing DNA damage.

We have made corresponding modifications on page 4, lines 126-131.

Comment 7: You must enter an explanation of the abbreviations: MDC1, BRCA1, ATM, ATR, ITGAV, TNBC, WGCNA etc.

Our Reply: We agree with this suggestion. We have added the explanation in the article.

Mediator of DNA damage checkpoint 1 (MDC1), page 6, line 207.

Breast and ovarian cancer susceptibility protein 1 (BRCA1), page 6, line 208.

Ataxia Telangiectasia Mutated (ATM), page 6, line 208.

Ataxia telangiectasia and Rad3-related protein (ATR), page 5, line 188.

Integrin alpha V (ITGAV), page 11, line 335.

Triple negative breast cancer (TNBC), page 12, line 379.

Weighted correlation network analysis (WGCNA), page 13, line 415.

General control non-derepressible 5 (Gcn5), page 7, line 252.

Comment 8: Lines 206-208

«Further research using protein-protein interaction analyses confirmed that the HMG box, a domain-specific to vertebrates and of recent evolutionary origin, facilitates interactions between WDHD1 and mismatch repair protein 2 (MSH2)»

HMGB-domain proteins are found in all eukaryotes and it is incorrect to say that "the HMG box, a domain-specific to vertebrates and of recent evolutionary origin". If the authors want to describe evolutionary differences in organisms, then it is worth introducing additional information on this topic.

Our Reply: We thank the reviewer for this valuable suggestion. We want to emphasize that the HMG box in WDHD1 is domain-specific to vertebrates. WDHD1 for invertebrates does not consist of the HMGB domain.

The sentence now reads: “Further research using protein-protein interaction analyses confirmed that the HMG box in WDHD1, a domain-specific to vertebrates WDHD1 and of recent evolutionary origin, facilitates interactions between WDHD1 and mismatch repair protein 2 (MSH2) .” Page 6, lines 216-218.

Comment 9: Lines 213-217

«In addition to its regulatory functions in DNA replication and DNA damage repair, recent research has revealed the involvement of WDHD1 in promoting mouse embryo development [18]. WDHD1 is important for post-implantation embryo development in mice. Furthermore, the knockdown of WDHD1 during larval development results in lethality [62]»

Here the authors write about post-implantation embryo development in mice. The concept of "larval development" is given incorrectly.

Our Reply: We thank the reviewer for this suggestion. The sentence now reads: “WDHD1 is important for post-implantation embryo development in mice and the knockdown of WDHD1 results in embryonic lethality.” Page 6, lines 225-226.

Comment 10: GCN or Gcn, CDC45 or Cdc45?

Our Reply: We thank the reviewer for this valuable suggestion. We have modified “GCN” to “Gcn”, “CDC45” to “Cdc45”. Page 5, line 152. Page 7, line 263.

Comment 11: Lines 273-274

respectively.

Our Reply: We thank the reviewer for this comment. The sentence now reads: “Figure 4A shows increased WDHD1 expression levels in various tumor types, including breast cancer (BRCA), esophageal carcinoma (ESCA), head and neck squamous cell carcinoma (HNSC), lung adenocarcinoma (LUAD), pancreatic adenocarcinoma (PAAD) and so on .” Page 9, lines 287-289.

Comment 12: Lines 299

Please clarify in the text: «specific HPV genotypes, such as 16 and 18», E7 and HPV E7.

Our Reply: We thank the reviewer for this valuable suggestion.

“Specific HPV genotypes” refers to high-risk HPV, which is the subtype of HPV that is more likely to cause cervical cancer. Page 11, line 317.

As an important viral oncogene in HPV, E7 can be expressed in HPV-infected cells and induce cervical cancer. Page 11, line 319.

Comment 13: Line 351

Please clarify in the text: «a nuclear interaction between WDHD1 and MAPRE2»

Our Reply: We thank the reviewer for this comment. The sentence now reads: “Gou et al. further revealed the upregulation of MAPRE2 following WDHD1 knockout and identified a protein-protein interaction between WDHD1 and MAPRE2 in nuclear”. Page 12, line 373.

Comment 14: Lines 395-396, 403-405

It is necessary to describe in more detail the effect of WDHD1 on the diagnosis and treatment of pancreatic adenocarcinoma disease and cervical cancer, underscoring its clinical relevance in the text.

Our Reply: We thank the reviewer for this valuable comment and the suggestion of two references. We have emphasized the clinical relevance of WDHD1 to pancreatic cancer and cervical cancer on page 13, lines 418-425.

Comment 15: Line 458-460

 «Analyses of dependency scores have revealed that WDHD1 has a negative dependency score in different types of cancer, highlighting its essential role in the growth and proliferation of pan-cancer cells»

It is necessary to clarify in the text: «WDHD1 has a negative dependency score in different types of cancer»

Our Reply: We thank the reviewer for this valuable suggestion.

The median dependency score of WDHD1 shown in the DepMap database (https://depmap.org/portal/) is lower than -1, revealing the essential role of WDHD1 in the growth and proliferation of pan-cancer cells.

DepMap database includes three large-scale projects (Broad Achilles, Sanger, and GeCKO) of CRISPR screening and has become an attractive and powerful technique to identify target genes essentiality data. Gene dependency scores in DepMap are normalized such that a value of 0 represents the median dependency score of negative control genes and −1 represents the median dependency score of sgRNAs targeting pan-essential genes.

We have made modifications on page 15, lines 491-493.

Comment 16: Figure 2

The legend should be more detailed: the information in the legends to the Figure is not enough to understand of the presented material.

HMG (panel C) and HMG box (panel B) must be replaced by a HMGB domain. Panel C: SepB and WD40 must be replaced by a SepB domain and WD40 domain respectively.

Our Reply: We agree with this suggestion. We have made modifications in Figure 2. Page 4.

Comment 17: Figure 4

Panel A - The labels on the axis X are not visible.

The information in the legends to the Figure is not enough to understand the presented material.

Our Reply: We thank the reviewer for this valuable comment.

The X-axis of Panel A in Figure 4 before has shown the group of tumor and normal and the sample size of each group. We decided to use color to distinguish tumor and normal group now, and each point represents a sample.

Figure 4A shows the differential expression levels of WDHD1 in tumor and normal tissues. Gene expression levels were determined by TPM value and the cancer type is shown above.

Figure 4B shows WDHD1 alteration frequency including mutation data and copy number alteration (CNV) data in various tumors. The cancer type is shown below. Green, mutation frequency; red, amplification frequency; blue, deep deletion frequency; grey, multiple alterations frequency.

Page 10.

Comment 18: Figure 3

The information in the legends to the Figure is not enough to understand of the presented material.

Our Reply: We thank the reviewer for this valuable comment. We have added some information to the legend to help understand.

Panel A: WDHD1 participates in DNA replication by forming replisome with Cdc45/Mcm2-7/GINS complex and polymerase α. When replisome is present, one parental DNA forms two progeny DNA.

Panel B: Ionizing radiation (IR) leads to double-strand DNA breaks (DSBs). WDHD1 binds to CtIP, recruiting Rad50, Rad51, NSB1, BRCA1, and participate in DNA homologous repair to repair DSBs.

Panel C: In the metaphase of mitosis, WDHD1 works in conjunction with HJURP to recruit CENP-A to format centromeres, making two sisters chromatid connect together.

Panel D: Combining WDHD1 with Gcn5 can avoid the ubiquitination degradation of Gcn5. When WDHD1 and Gcn5 are isolated, Gcn5 will be ubiquitylated and degraded by the Cdt2-DDB1-Cul4A/B-Roc complex. E2 represents a ubiquitin-conjugating enzyme, and U represents ubiquitin.

Page 7.

Comment 19: References

№ 30. «doi:Doi» must be replaced by «doi»

Our Reply: We thank the reviewer for this suggestion. «doi:Doi» has been replaced by «doi» on page 20, line 656.

We thank the anonymous reviewers again for their constructive comments. All modifications have been marked in red.

Round 2

Reviewer 2 Report

Dear authors, thank you for your comments.

The article can be accepted for publication in the «International Journal of Molecular Sciences» with minor correction.

Comment 3: Lines 77-78

«The human WDHD1 protein is considered the homologous gene of the budding yeast CTF4.»

It seems to me that here it is necessary to replace “The humam WDHD1 protein” by “The human  WDHD1 gene.